# Rights of Indigenous Children: Reading Children's Literature through an Indigenous Knowledges Lens

**Shelley Stagg Peterson * and Red Bear Robinson**

Department of Curriculum, Teaching and Learning, University of Toronto, Toronto, ON M5S 1V6, Canada;
redbear.robinson@mail.utoronto.ca

* Correspondence: shelleystagg.peterson@utoronto.ca

**Abstract:** Indigenous children's literature supports Indigenous communities' rights to revitalization, and to the transmission to future generations, of Indigenous histories, languages, and world views, as put forth in the United Nations Declaration of the Rights of Indigenous Peoples. Drawing on Indigenous teachings that were given to him by Elders, an Indigenous Knowledge Keeper, Red Bear, interprets 10 Indigenous picture books published in Canada between 2015 and 2019 by mainstream and Indigenous publishing companies. These books were selected from the International Best Books for Children Canada's list of Indigenous books and websites of four Canadian Indigenous publishers. We discuss the Knowledge Keeper's interpretation of books that are grouped within four categories: intergenerational impact of residential schools, stories using spiritual lessons from nature, autobiography and biography, and stories using teachings about relationships. Recognizing the richness, authenticity, and integrity of Red Bear's interpretation of the books, we propose that all teachers should strive to learn Indigenous cultural perspectives and knowledge when reading Indigenous children's literature.

**Keywords:** Indigenous children's literature; Indigenous knowledges; Indigenous children's rights; cultural authenticity

## 1. Introduction

Redressing Canada's long history of assimilative practices denying Indigenous children's rights requires action toward reconciliation in all realms of society, but especially in education [1]. The first principle for reconciliation is that Indigenous children have a right to an education that includes "the foundations of their own knowledge systems and that enables them to benefit from the education in their lives" [2] (p. 77). We propose that integrating Indigenous children's literature into classroom practice, along with adopting Indigenous knowledge as a framework for reading the literature, are promising tools aligning with this principle for reconciliation. Cultural authenticity of picture books used in classrooms, and of the reader's interpretation of the literature, are both important to support children's development of strong cultural identities. We define cultural authenticity in terms of cultural perspectives, values, and experiences of the people represented in the book [3].

Our proposal is based on a recognition that children's literature is a "means by which society transmits selective cultural understanding" [4] (p. 183). Indigenous authors, illustrators, and publishers challenge Eurocentric notions of contemporary and historical Indigenous worldviews, experiences, and values, as well as relationships between Indigenous and non-Indigenous peoples in Canada. Indigenous children's literature supports Indigenous children's families' and communities' rights to "revitalize, use, develop and transmit to future generations their histories, languages, oral traditions, philosophies, writing systems and literatures" as detailed in the United Nations Declaration on the Rights of Indigenous Peoples Article 13 [5].

The purpose of our paper is to highlight the cultural authenticity of 10 recently published Indigenous picture books written in the English language by Indigenous authors. Red Bear, an Indigenous Knowledge Keeper, draws on teachings from Elders in his community and his own experiences to interpret the books. We begin with a brief description of the context of children's literature publishing in Canada.

## 1.1. Indigenous Children's Literature in Canada

In Canada, the publishing of Indigenous authors' picture books has grown over the past decade. At the time of writing this paper, trade books for children are published by three Indigenous publishers located across the country—in two of Canada's ten provinces and one of its three territories. Non-Indigenous publishers of children's literature, domestic or part of international companies (e.g., Second Story Press in Toronto, Ontario), are also publishing children's literature created by Indigenous authors.

Resisting mainstream commercial inclinations, Indigenous publishers of children's literature are located outside Toronto and Vancouver, the major urban centers that house non-Indigenous book publishers in Canada. We list them below in order of the year in which they were established.

1. Theytus Books, one of the oldest Indigenous publishing houses in Canada, is First-Nations-owned and located in Syilx territory on the Penticton First Nation Reserve in British Columbia. In Salish, "theytus" means "preserving for the sake of handing down", a practice that Theytus Books has been carrying out toward the goal of publishing books with "the highest level of cultural authenticity and integrity" [6] since 1980. Theytus Books uses oral tradition practices and consults Elders and traditional leaders throughout its publication process. Publishing practices respect Indigenous protocols regarding the appropriateness of representing certain cultural aspects in literature [7].

2. Inhabit Media, the first Inuit-owned, independent publishing company in the Canadian Arctic, is located in Iqaluit, Nunavut. Publishers brought Inhabit Media into being in 2006 to:

   > … promote and preserve the stories, knowledge, and talent of the Arctic, while also supporting research in Inuit mythology and the traditional Inuit knowledge of Nunavummiut (residents of Nunavut, Canada's northernmost territory) … Our authors, storytellers, and artists bring traditional knowledge to life in a way that is accessible to readers both familiar and unfamiliar with Inuit culture and traditions. [8]

   > With the goal of ensuring that Nunavut children see their lives and culture represented authentically and accurately in children's literature, Inhabit Media creates books that are "reflective of everyone, the culture, the lifestyle, the sensibilities and the values of the place in which we live". [9] (p. 15)

3. Kegedonce Press, owned by the Chippewas of Nawash First Nation and located in Neyaashiinigmiing, Ontario, began in 1993. Kegedonce Press strives "to foster the creative cultural expression of Indigenous Peoples through the publication of beautifully crafted books which involve Indigenous Peoples" [10]. HighWater Press, an imprint of Portage & Main Press in Winnipeg, Manitoba, publishes books that "portray a wide-ranging express of Indigenous peoples' culture and experiences" [11].

Indigenous authors', illustrators' and publishers' high quality and growing body of work sends a strong message that Indigenous ways of being in and seeing the world are valued. These books also counteract the longstanding practice of appropriation of Indigenous stories by non-Indigenous writers of children's literature [7]. Contemporary Indigenous children's literature, created and published by Indigenous creative teams, is playing a role in "reclaiming, revitalizing, and sustaining Native spirituality, world view, culture, and literacy" [12] (p. 14). Indigenous authors and illustrators write

for Indigenous children, imagining readers who bring Indigenous perspectives and experiences within a colonialist society to their reading [7]. Indigenous children can relate to Indigenous characters who live familiar lives and view the world from familiar perspectives [13].

Indigenous authors, illustrators, and publishers introduce new ways of thinking about narrative structures and features, such as Indigenous perspectives on the role of spiritual and natural worlds, to children's literature. These structures and features are being recognized, as Indigenous children's literature is winning prestigious national awards. Two books have won the two top awards in the country in the past five years. Melanie Florence's *Nimâmâ*, (2015) [14] won the TD Canadian Children's Literature Award and David A. Robertson's *When We Were Alone* (2015) [15] won the Governor General's Award for Children's Literature—Illustration.

This recognition is long overdue in a settler country where Indigenous content and perspectives have been absent throughout most of the history of publishing children's literature in Canada. Indigenous writers have simply not had access to the publishing world [16]. As a result, more often than not, books published about Indigenous peoples have been written and illustrated, as well as formatted and marketed, from a non-Indigenous perspective. In many cases, these non-Indigenous authors have been "oblivious to the historical and symbolic processes that have privileged whiteness as a normative mode of being" [7] (p. 226), resulting in unconscious reproduction of stereotypes and misconceptions, particularly in representations of contemporary lives of Indigenous children and their families.

The presence of Indigenous publishers, authors, and illustrators in Canadian children's literature holds the promise that the worldviews and experiences of Indigenous children and their families and communities will be honestly represented [16]. These books are more likely than those created by non-Indigenous authors and illustrators to be authentic, as assessed using criteria proposed within the multicultural children's literature world: authentic author's perspective and experience vis-à-vis Indigenous culture and experiences, respectful portrayal of Indigenous languages and dialects, recognition of diversity across Indigenous peoples, and respect for Indigenous peoples' resilience and resistance of colonizing practices of non-Indigenous peoples [17,18]. For Indigenous children, participating in shared reading of Indigenous children's literature that draws on Indigenous knowledges is integral to supporting children's rights to "healing, self-determination, and reclamation of identity, language, and cultures" [12] (p. 10). For non-Indigenous children and adults, reading authentic Indigenous children's literature can lead to recognition of inequities and to creating the respectful relationships that are necessary for reconciliation [1].

Awareness of the literature and use of the criteria for assessing cultural authenticity, on the part of teachers, parents, and other adults in Indigenous and non-Indigenous children's lives, is a first step. Supported by research showing that non-Indigenous teachers tend to have limited knowledge of Indigenous culture, knowledge, and history [19,20], the reproduction of stereotypes can also occur when readers have an underdeveloped understanding of Indigenous cultures. Non-Indigenous readers show respect for Indigenous worldviews and knowledges by recognizing that Eurocentric, mainstream philosophies are not sufficient when reading Indigenous children's literature [21]. In order to build "intercultural understanding, empathy and mutual respect" necessary for reconciliation [1] (p. 7), non-Indigenous Canadians must begin to integrate Indigenous perspectives of historical and contemporary relationships between Indigenous peoples and other Canadians into readings of all literature, but especially of Indigenous children's literature.

*1.2. Introducing Indigenous Children's Literature Is Not Enough*

Practices for engaging children with literature should work toward developing "an appreciation for the sophistication of Indigenous knowledge and for the appropriate Indigenous pedagogies through which such knowledge can be explored" [21] (p. 820). The cultural readings of the selected picture books by Anishnaabe Knowledge Keeper Red Bear are drawn from the traditional teachings of respected Elders from his community. He is known as a Knowledge Keeper, whose traditional

knowledge is valued by his Indigenous community. His readings are in keeping with a belief that "Indigenous texts deserve to be read in the light of the cultures in which they are produced, and with due attention to their difference from Western texts, rather than from within the assumptions of Western culture and textual practices" [7] (p. 227).

It is important to "find the First Nations community leaders in your region, enter into a relationship with them, and learn their way of knowing the world" [22] (p. 329). We realize that Indigenous Knowledge Keepers cannot be present every time a non-Indigenous teacher reads an Indigenous picture book with children. However, we suggest that teachers create funds of knowledge about Indigenous experiences and perspectives to inform their readings of Indigenous children's literature. This knowledge can be created through consulting with Indigenous community members; websites created by Indigenous peoples about teachings and personal stories [23,24]; and literature written, illustrated, and published by Indigenous creative people in their countries. We hope that our paper, with readings of a number of recently published Indigenous pieces of literature by an Indigenous Knowledge Keeper, will enhance teachers' funds of knowledge, as well.

## 2. Materials and Method

### 2.1. Materials: Selection of Indigenous Children's Literature

Recognizing that the picture book is the most common form of children's literature [25,26], we selected 10 picture books from the body of Indigenous children's literature published between 2015 and 2019 by Canadian publishers. We used a publication from the International Best Books for Children Canada [27] to guide our selection of picture books published between 2015 and 2017. This annotated bibliography of 100 Indigenous children's books (from board books to picture books for older children) is the culmination of a project initiated in 2016 in response to the Truth and Reconciliation Commission's Calls to Action. The project's goal was to support Canadian Indigenous children's right to access to high-quality books that reflect their lives and allow the children to see themselves in the books. To add books published in 2018 and 2019 to our set, we consulted Indigenous publishers' websites and the fall 2019 issue of *Book News*, a magazine published by the Canadian Children's Book Centre. These books are available through Goodminds.com, an online distributor of books for children and young adults written and published by Indigenous book creators [28].

From these lists of books, we selected 10 books. Among the selection criteria are picture books with publication dates between 2015 and 2019, and authors who are Indigenous. Additionally, the books were assessed by Red Bear as being culturally authentic, based on his experience and teachings that had been given to him.

### 2.2. Method: Analysis of an Indigenous Knowledges Approach

We started our categorization of the books using Bradford's (2007) scheme [7]. She identified contemporary realistic fiction as the predominant genre, with retellings of traditional narratives as the second most commonly published genre/text type. Our categorization of books published approximately a decade later is more refined. We believe that the greater abundance of Indigenous picture books that are written and often illustrated by Indigenous authors and illustrators makes it possible to use content and themes, as well as genre/text type, to categorize the books.

We begin our description of the Indigenous Knowledges perspective guiding our interpretations of the 10 picture books by acknowledging that Indigenous knowledges are contextual and vary across Indigenous peoples. We do not present our perspective as a universal view of Indigenous knowledges. Generally, within Indigenous knowledge traditions, introductions to knowledge sharing start with each participant locating themselves geographically and in terms of their communities. Accordingly, Red Bear locates himself as of the Anishnaabeq Bear Clan, who is a Knowledge Keeper and Anishnaabemowin language teacher from Bawaating, the land of sparkling waters, now called Sault Ste. Marie, Ontario. He is now a PhD student in a southern Ontario university. Shelley is a settler

former elementary teacher with a background of family members from the Netherlands, Scotland, and Ireland. Her childhood and classroom teaching were in rural western Canadian schools where Indigenous and non-Indigenous children were her students. She is now a literacy professor in an urban Canadian university's faculty of education.

Indigenous knowledges generally start with the assumption that the physical and spiritual world (seen as broader than a religious spirituality) are interconnected. The spiritual realm is accessed through "personal and collective engagements at the level of intuition, meditation, prayer, ceremony, dreams, and vision quests, and other forms of introspection that reach another realm" [2] (p. 76). The inner knowledge that comes from connecting the affective, cognitive, and physical aspects of learning within the spiritual and natural world is at the core of Indigenous knowledges [2,29]. Land, as a "spiritually fluid, dynamic, and relational consciousness" [30] (p. 42), and relationships to Land are at the core of Indigenous knowledges. As expressed by Elders to Jeannette Armstrong [31], "it is land that holds all knowledge of life and death and is a constant teacher" [31] (p. 178). It is important to listen intently to the teachings of the land and then to retell its stories to our succeeding generations. The teachings are contextualized, experience-based, and remembered within our body, mind, and spirit. Embodied knowledges, often referred to as body memory, are connected to Land through the teachings and the stories through which they are transmitted [32].

The wholistic Indigenous land-based knowledges approach that Red Bear has taken up in his interpretation of the 10 books differs from the non-Indigenous approaches to reading and discussing multicultural picture books that are more widely employed in analyses of multicultural literature. For example, a critical literacy approach [33,34] involves taking up multiple viewpoints to disrupt commonly held assumptions that perpetuate inequities and oppression with a goal toward promoting social justice. Taking the perspective of all who play a role in the narratives and issues of children's literature, children and teachers identify power relationships and misconceptions in discussions of children's literature.

Another example of a widely used non-Indigenous approach is a critical reader's response approach [35,36], which is characterized by conversations about picture books that draw from their personal experiences, perspectives, and emotional responses to construct meanings related to social justice and racism. These meanings can be on a continuum from very personal, as children create aesthetic meanings, or less personal, as children create efferent meanings [37]. Although the critical reader's response approach does draw on the emotional, it and the critical literacy approach are heavily steeped in intellectual activity and in Eurocentric worldviews. They do attend to connections between social action and the intellectual, but are not as holistic as what we are proposing as an Indigenous Knowledges approach, in connecting social action and the spiritual and natural world [38,39].

After an initial reading of the 10 picture books at his home, Red Bear joined the other author in her office. He read the pictures and text of each of the 10 picture books, voicing his thoughts as he read. The other author wrote verbatim what he said, asking for clarification when she was unsure of the meaning of his voiced interpretations. Red Bear talked about the authentic representation in illustrations, as well as illustrations that misrepresented Indigenous symbols and meanings, as some of the illustrators were non-Indigenous.

We acknowledge that there is no universal Indigenous perspective and that Red Bear's interpretations reflect the worldviews and experiences of his region. To gain a more regionally accurate interpretation of books that are set in other regions of Canada, we would have to interview the authors and the Indigenous peoples of the communities represented in the books.

## 3. An Indigenous Knowledge Keeper's Reading of Books through an Indigenous Knowledges Lens

Using inductive methods, we found that the 10 books could be grouped into three categories; two categories are based on the themes within the stories and one is the story type. The themes within the books in the latter category, autobiography and biography, contain the themes of the other two categories. We use it as a separate category to highlight how some of the Indigenous picture books are

based on stories of the lived experiences of individuals and others are based on authors' creation of characters whose experiences are a composite of many people who authors have met or are part of their families. We report Red Bear's interpretation of the books, using an Indigenous Knowledges approach, in terms of these categories:

1.  Non-Autobiographical/Biographical Narrative: Theme of intergenerational impact of residential schools
2.  Non-Autobiographical/Biographical Narrative: Theme of using spiritual lessons from nature
3.  Autobiography and biography: Themes of intergenerational impact of residential schools and spiritual lessons from nature

A common thread across our categories, as shown in the following discussion of our findings, is relationship. Heath Justice [29] explains that relationships are the central theme of Indigenous literature: relationships to the land, to self, and to others (e.g., ancestors, descendants, local and broader communities, and to the spiritual world), as well as to histories and futures.

Please note that in the following reporting of Red Bear's interpretations, we attempted to identify the affiliation of authors and illustrators of all of the books because it is important to acknowledge where knowledge embedded in the books comes from [40]. In some cases, we were unable to find this information.

You will find bibliographic information about all books in Figure 1.

*When We were Alone* – author David Robertson, illustrator Julie Flett. Published by Highwater Press, Winnipeg, Manitoba, 2015.
*Stolen Words* – author Melanie Florence, illustrator Gabrielle Grimard. Published by Second Story Press.
*Keeshig and the Ojibwe Pterodactyls* – story told by Keeshig Spade to Celeste Pedri-Spade, illustrators Robert Spade and Kiniw Spade. Published by Kegedonce Books, 2018.
*Sukaq and the Raven* – authors Kerry McCluskey and Roy Goose, illustrator Soveon Kim. Published by Inhabit Media, 2017.
*The Girl and the Wolf* – author Katherena Vermette, illustrator Julie Flett. Published by Theytus Books, 2019.
*When the Trees Crackle with Cold (A Cree Calendar)* – author Bernice Johnson-Laxdal, illustrator Miriam Körner. Published by Your Nickel's Worth, Regina, SK, 2016.
*In my Anaanas' Amautik* – author Nadia Sammurtok, illustrator Lenny Lishchenko. Published by Inhabit Media, 2019.
*Meennnunyakaa Blueberry Patch* – author and illustrator Jennifer Leason; author and translator Elder Normand Chartrand. Published by Theytus Books, 2019.
*The Water Walker* – author and illustrator Joanne Robertson. Published by Second Story Press, 2017.
*I am Not a Number* – authors Jenny Kay Dupuis and Kathy Kacer, illustrator Gillian Newland. Published by Second Story Press, 2016.

**Figure 1.** Children's Literature Cited.

### 3.1. Intergenerational Impact of Residential Schools

In Canada, the traumatic abuse experienced by Indigenous children and their families has had ongoing effects on the lives of Indigenous children today [2]. The residential school policy was implemented by the Canadian government with, in the words of the first Indian agent, Duncan Campbell Scott, the goal of "killing the Indian in the child" [41]. Although the stories of individual survivors are very traumatic, Red Bear explains that they need to be shared through intergenerational storytelling and picture books created by Indigenous authors who have firsthand experience or pass down stories from their parents and grandparents. This sharing can lead to better understanding, on the part of Indigenous and non-Indigenous children, of the intergenerational impact of these horrific assimilative practices. Among the 10 picture books in our selection, Indigenous authors, and in some cases Indigenous illustrators and publishers, have created honest, sensitive, and powerful

picture books whose contributions to reconciliation can be amplified when read through an Indigenous Knowledges lens.

An example of a book that tells the story of an individual survivor of residential schools is *When We Were Alone.* The author, David A. Robertson, is a member of Norway House Cree Nation and the illustrator, Julie Flett, is Cree-Métis. Both are from Manitoba. Connections to nature and the sense of being part of the natural world abound in the parts of the book where Kôkum is an adult talking with her granddaughter and in the remembrances of her time at a residential school when she and her sister were alone beyond the reach of the residential school educators and administrators. When Kôkum and her sister were alone, they could regain their oneness with nature by covering themselves with colored leaves of the fall and lengthening their hair with long blades of grass. Illustrations of Kôkum as an adult show her colorful flower-patterned dress blending in with the flowers growing in the bush. Her long hair imitates the vines. When the two sisters were alone, they used nature to retain their teachings and their Indigenous identity. The traditional jingle-dress dancer sounds and moves like the birds flying through the air, showing that Indigenous teachings and healing ceremonies imitate nature. Traditional beliefs in animals being able to speak Indigenous languages are reflected in the image of a bird flying amidst the writings of the Cree language. These ways in which Kôkum and her sister blend in with nature and show that they are not greater nor lesser than anything in nature contrast with the identities imposed upon them in the residential school.

The comparisons of the residential uniforms with storm clouds and juxtapositions of cuttings of the children's hair with blades of dead grass show that the residential school practices went against Indigenous teachings and ways of living. Drab colors and rigid lines of illustrations portraying the children while in residential schools give a sense of conformity and rigidity. The Indigenous children were not allowed to have their own world views, language, and ways. The sweet singing of the single bird when the children were alone is replaced with the raucous cawing of crows when referring to the sounds of English speech that Indigenous children were forced to speak in the residential schools. Indigenous teachings portray crows as very clannish and garrulous, sometimes considered good qualities and other times as undesirable qualities. As children, Kôkum and her sister recognized the importance of continuing to speak their language to maintain their Indigenous identities. The two girls used every opportunity to come together to speak Cree. Because of the draconian punishments inflicted on Indigenous children who spoke their own languages, the girls had a sense that they had to whisper the words and be wary of being caught when they came together to hold hands and use their language. Their Cree voices and their family relationships had to be hidden, as children were punished when they were seen in the company of their siblings in the residential schools. The image of the girl alone in a room looking out the window at nature portrays that lonely existence.

A second book that helps us to understand the intergenerational impact of residential schools, *Stolen Words*, is written by Cree-Métis author, Melanie Florence. She tells this story in honor of her grandfather, who attended residential schools in western Canada. This story of a seven-year-old girl's grandfather, who is a survivor of a residential school, reflects what many Indigenous people are still going through today in Canada. Grandfather is living with the effects of having his language, culture, and self-respect stolen from him. He shares his feelings and experiences with his granddaughter.

The image of the blackbird/crow/raven (it is difficult to tell which black bird is represented), formed from the flow of children's Indigenous words, pictured as a remembrance of the grandfather's time in residential schools, gives a spiritual sense to the children's Indigenous language. The cage was too small, yet the blackbird/crow/raven was forced to fly into it. Although the punishments and shaming in residential schools broke the spirit of children who grew up to be men like the girl's grandfather (and many never regained their language and culture), the cage could not hold the blackbird/crow/raven forever. The new generation is picking up the language and helping residential school survivors regain their language, culture, and self-respect. Indigenous teachings tell us that everybody in creation (babies, children, adults, Elders) can teach us life lessons. To Grandfather, the words, which are released as blackbirds/crows/ravens flying from a cage on the page, feel like home and mother. The two are

viewed as caregivers: home is the place and mother is the person who makes you safe and feel loved, and also helps you to live in a good way.

There is some evidence of misconceptions of Indigenous traditions in the illustrations by non-Indigenous illustrator, Gabrielle Grimard. For example, the illustration that includes a dreamcatcher, which is revered as a protector when we sleep, seems to be a prop or decoration. The dreamcatcher captures the bad spirits that would make us sick or harm us in some other way, entering through our dreams. However, the dreamcatcher in the girl's hand seems to have been commercialized with plastic beads and strings. It is not made of willow and sinew, which come from a tree and an animal. All the teachings, including acknowledging that the tree and animal have given up something of themselves, are lost. In addition, the dreamcatcher has many feathers, which is unusual for a child's dreamcatcher. The feathers of the dreamcatcher have to be earned, and the teachings of the feathers are given by an Elder or Knowledge Keeper to the child when she is making the dreamcatcher. Additionally, in some Indigenous communities, the blackbird, the crow, and the raven are tricksters who provide teachings by playing tricks on people to get them back on track when they are heading in a harmful direction. Red Bear does not see how this symbolic attribute of the blackbird/crow/raven fits with the theme of languages being stolen from children while in residential schools. What he sees as a result of the loss of Indigenous language and culture is the picking up of artifacts of traditional ways without learning the teachings that make them so important culturally. Given that the illustrator is not Indigenous, we are not certain that she was aware of this when creating the illustrations.

### 3.2. Stories Using Spiritual Teachings from Nature and Relationships

Red Bear explains that every element of the natural world has a teaching that can be woven into a story. Stories based on these teachings recognize the interconnections between the physical and spiritual, and show us how to live in a good way. In these stories, we come to recognize that "the land is a gift given to us by the Creator. By acknowledging the land in this way, we affirm our relationship with all of its beings" [32] (p. 62). These stories also teach about the four stages of the circle of life: infant, youth, adult, and elder. Drawn from many different sources, particularly the Seven Grandfather Teachings: wisdom, love, respect, bravery, honesty, humility, and truth [42], the teachings explain roles and responsibilities at each part of the life circle. Many stories transmit the wisdom of the land, interpreted through many generations of storytelling, to new generations.

An example of an intergenerational story is *Keeshig and the Ojibwe Pterodactyls*. Seven-year-old Keeshig tells of the spirit, Nanaboozhoo, within a mountain, also known as the Sleeping Giant, on the shores of Lake Superior, and his relationship with the Thunder Birds (also known by Keeshig as Ojibwe pterodactyls). Red Bear knows of many sacred ceremonies that are held at Mt. McKay, which is part of the Sleeping Giant. Keeshig's mother is the co-author, who engages in Indigenous teaching practices by asking questions to deepen his understanding and encourage Keeshig's imagination.

The conversation takes place after the family has been dancing traditional dances at a pow wow, which is a social gathering with traditional drums and dancing, in which many nations come together to celebrate a wide range of seasonal changes. Keeshig's mother is wearing a jingle dress, which is significant during the healing dance ceremony. The dress usually has 365 cones; one for each day of the year. She would have attached one cone each day of the previous year to make the dress. These cones jingle when she dances in time with the drum, which represents the heartbeat of the earth. The brilliant colors reflect the vibrant colors of Mother Earth. Keeshig integrates the teachings of his family and community (e.g., the Thunder Birds are the spirits of the eagles after leaving this world and entering the spirit world) with his own story about Ojibwe pterodactyls. They eat what the eagles would have eaten while on earth. The illustrations are adult-created up to the point where Keeshig says that the Ojibwe pterodactyls' wing beatings are Nanaboozhoo's heartbeats. The drawing for this part of the story is a child's drawing of a child-imagined teaching. Keeshig's mother acknowledges her son's teaching by calling the children her heartbeat. Like all things that are good, her son's love is good medicine.

Red Bear contextualizes young Keeshig's story by drawing on his decades of experience in this northern region of Canada. The story, itself, however, is unusual, in terms of the intergenerational storytelling coming from the imagination of a young boy, who shares his insights and observations with adult family members, rather than the other way around.

Just as author Roy Goose passes on the teachings he learned from his great-grandmother, Naimee Mammayuk, to his grandchildren, the creation story in *Sukaq and the Raven* is told by a grandmother to her grandson. The story is from Inuvialuit oral culture in Nunavut. It has much in common with the teachings communicated through stories, within Red Bear's Anishnaabe experience.

Red Bear explains that Indigenous creation stories tend to have an animal trickster, who must create the earth out of necessity. In this Inuvialuit story, the raven, a trickster, needed a place to land. The creation story reflects the geography of Nunavut: snow gathered on the biggest-ever raven's wings, forming a snowball that becomes so large, the raven could land on it. Pecking in the snow with its beak, the raven brought plants to earth, which had lights that became the sun and moon when he needed light. Plants also had the form of a woman in them, when he needed companionship. He breathed life into her. As in many Indigenous creator stories, living things came from the earth. Like tricksters that Red Bear is familiar with in other Indigenous cultures, the raven is able to take the form of a man. Animals are created after man is created. Indigenous stories of the creation of animals usually start with animals that are important within the local environment (e.g., in Red Bear's Ontario location, the story of the creation of the bear comes first, followed by the crane, the loon, birds, fish, deer, and the marten).

In Indigenous culture, the dream world has many dimensions, including the conscious/awake state, the almost-asleep state, and the dream world. Red Bear notes that the illustrations are consistent with these cultural understandings of dreams. They show Sukaq rising out of his bed through these states of consciousness as his anaana (mother) tells the story.

Katherena Vermette, a Métis writer from Treaty One territory in Manitoba, wrote *The Girl and the Wolf*. Julie Flett, illustrator of *When We Were Alone*, illustrated this book, as well. Although the author explains that this is a totally made-up story, Red Bear feels that there are plenty of elements that she must have experienced or heard from family and community members, as Indigenous teachings are evident throughout the story. The story starts with an experience that is common to many Indigenous families: camping out in the bush to pick berries together. The little girl ventures away from her family, getting lost. A wolf (Red Bear expects that the wolf is most likely the young girl's clan or dodem) appears from between some trees. The animal of one's clan is a spiritual helper and teacher who teaches the attributes of the clan. The wolf is very family-oriented. Wolves mate for life and both males and females look after their young, teaching them how to survive.

In the story, the wolf teaches the little girl that she could help herself if she calmed down, closing her eyes and looking inside herself. The wolf knows that she has the knowledge within her from the teachings of her family. Red Bear explains that from an Indigenous perspective, the knowledge is inside us if we seek it. For example, the girl knew that in order to satiate her hunger, she could eat the berries by the water, which are not poisonous, although those deeper in the bush might be. When the wolf guided her to look around, she noticed the skinny trees where her family camped, and was able to find her way back to them. The girl told her story to her family and that evening, tied tobacco in red cloth, and left it for the wolf to say thank you. She made a tobacco bundle, which Red Bear explains, is a traditional way to say thank you. Red Bear explains that this story, like many Indigenous teachings for children, involved troublesome happenings occurring when children do not listen to parents. However, it also shows that the animal representing our dodem is always there to guide us, even when we cannot see it.

The childhood of author Bernice Johnson-Laxdal, a Cree language and culture teacher from Ile-a-la-Crosse, Saskatchewan, involved traditional activities that are dependent on the seasons of the year. She shares these activities in *When the Trees Crackle with Cold (A Cree Calendar)*. Reflecting traditional knowledge of the seasonal cycle, the Cree calendar consists of 12 moons within six seasons.

Red Bear explains that the moons have different names, depending on where the Indigenous community is located in Canada, because the names reflect what is happening in nature in their particular locations during that cycle of the moon. There is a natural balance between nature and humans, with nature signaling the activities that people should be engaged in for survival. For example, when the young birds practice using their wings in the time of the Flying-Up Moon, people are picking blueberries in the old burn. Red Bear observes that the primacy of nature is reflected in the birds, animals, and plants being represented as far larger, in relation to people, than they would be in the natural world.

Nadia Sammurtok, an Inuit writer from Rankin Inlet, Nunavut, writes about one of the Grandfather Teachings—Love, in the picture book, *In my Anaanas' Amautik*. She compares a mother's love for her infant to the love of Mother Earth for all living things. The infant is carried in the hood or amautik of the mother's parka. This Inuvaluit story reminds Red Bear of stories he heard as a child about how babies feel in a tikinagan ("tik" means tree or wood and "nagaan" means vessel—it is a cradle board in which the baby is placed while wrapped in a moss bag to provide warmth, comfort and security). Relationships between mother and child are compared to relationships with the natural world. Mother Earth provides all that we need and is a beautiful environment for us to meet our needs. The baby compares the security feeling of the hood that is reminiscent of the iglu home, which protects the baby from the elements. Placing the baby in a safe place close to running water is a common practice to soothe the baby. Red Bear remembers hearing his grandmother saying that a mother's kisses are like the wind caressing the baby's face. The baby uses all senses to describe what it feels like in her mother's amautik. Subtle colors and soft lines of the illustrations speak to the calmness of the baby's relationship to their mother and to nature.

*3.3. Autobiography and Biography*

It is important for Indigenous children to see their own lives reflected in autobiographical stories of traditional family activities (Heath Justice) [29]. Indigenous authors of the stories in the books in this category narrate events from their childhood that continue to influence their lives today. Many teachings are embedded within these family stories. They are a way of transmitting culture and skills to children.

Jennifer Leason, a Saulteaux-Métis Anishinaabek member of the Pine Creek Indian Band, is from Duck Bay, Manitoba, where the story *Meennnunyakaa Blueberry Patch* is set. She is the great-niece of Normand Chartrand, also a Saulteaux-Métis Anishinaabek member of the Pine Creek Indian Band, who translated this dual-language Cree/English book. Like the narrator—an Elder who tells of his family going blueberry picking when he was young—Red Bear has fond memories of setting up a camp with other families from his community to pick blueberries, and, in the case of Red Bear, apples. This was also the time to go hunting for crow eggs. Travel was by horse and wagon or on foot. Horses are not part of Indigenous teachings because they were introduced to this continent. There is great respect for what nature supplies in order for humans to survive and be comfortable over the winter. Everyone shares, as is expected in Indigenous culture, and the communities camp together, staying until the blueberries have all been picked. The Elder as a child already had much knowledge about wildlife and nature, and about the importance of nurturing body, mind, and spirit. His telling of the story reflects senses that are alive to what nature offers: the water, fragrance of cut grass, and the burning of sweetgrass to smudge/cleanse. The illustrations contribute to this sense, as they use vibrantly colored organic forms to represent the life in leaves, flowers, birds, and other living things.

Biographies are also important for showing selfless and heroic acts of Indigenous individuals, like Josephine Mandamin, who have made a difference in the world in contemporary times. Author of *The Water Walker*, Joanne Robertson is a member of Atikameksheng Anishnawbek First Nation. She writes about an Ojibwe woman, Josephine Mandamin Mandamin (which means "corn" in Anishnaabemowin), who walked around all the Great Lakes while in her 70s. A bawagaajgan, or place where the truth is always told, comes from the place of dreams between being awake and the spirit world. In this place,

Josephine met an ogimaa, who pointed out that someday, all peoples must recognize the value of water. Illustrations show a number of ways in which people today disrespect and waste water, "making it unfit for life".

Josephine is portrayed with her copper water pail (Red Bear explains that copper has been used by Indigenous peoples because of its healing properties). She was the Water Walker who loved water (nibi) and water loved her. Throughout the book, the reciprocal relationship between water and humans is highlighted (e.g., we need it to wash, to drink, for transportation, for fishing for food, for enjoying and relaxing, etc.). Water, as explained by Red Bear, is used in tea and other foods, which are considered medicines. The overall message is that it is important to respect water and all that it gives us.

For seven years, she carried the Migizi (eagle) staff to lead fellow water walkers around the Great Lakes. The Migizi staff is a very strong protector of all of nature that is carried to all traditional ceremonies of the Anishnaabeg. In this case, the Migizi is meant to protect water. Josephine and her fellow water walkers left semaa (tobacco), which Red Bear explains is one of four sacred medicines (cedar, sage, tobacco, and sweetgrass) in every body of water they encountered. The Anishnaabe names of the Great Lakes (gichigami) describe what the lakes do (e.g., Niiganani-gichigami, as Red Bear explains, is the lake that leads/is out in front as it empties into the St. Lawrence River). The salt water poured into the lake represents the tears that Mother Earth is shedding over the polluting of the lakes. Josephine never stopped trying to protect the water, getting others to join her from across Turtle Island (North America).

Like the fictional narratives in which residential school survivors tell stories of experiences in residential schools, biographies, such as *I am Not a Number*, are important to ensure that settler and Indigenous children do not forget how Canadian government policies for all Indigenous children at one point in the country's history had an enormous and devastating impact on individual children. Anishnaabe author Jenny Kay Dupuis, from Nipissing First Nation in Ontario, collaborates with non-Indigenous author Kathy Kacer to tell the story of Jenny Kay's grandmother and her family during the time of enforced attendance at residential schools. Irene Couchie's father was the chief of their First Nation, yet he was not shown the respect that his stature should have been accorded when he confronted the Indian agent who took away his three children. The Indian agent, portrayed as a self-righteous and domineering giant figure representing the powerful Canadian government, blocked the doorway as Irene's father protects his children and wife by standing in front of them. Like Irene, Red Bear felt a sense of resentment and bewilderment about how his parents could allow him to be abused in the schools.

Irene vowed never to forget who she was, despite being told by a nun that she was only to be known by her number and not by her name. Red Bear explains that unlike the ceremony of being given a name in Indigenous culture, this number was randomly assigned by a sneering nun. In Irene's community, her parents would have given tobacco, a sacred medicine, to an Elder. The Elder would dream the child's spirit name, and then give the name to the baby in a ceremony. Family and community members attending the ceremony would say the name to the baby in each of the four directions (east, south, west, north). The images of the child scrubbing all the brown off her skin, having her hair cut, and wearing, like all the other girls, drab gray clothing, illustrates the horrific residential school goal of assimilation. The children were punished just for being themselves—for speaking their language, which a nun demonized as "the devil's language." As Red Bear explains, the cruel lessons which took the forms of being forced to hold a bedwarmer taken from the coals of a stove and memorizing songs or mending clothes, are in stark contrast to the teachings that Irene would have learned through stories and experiences in nature at home with her family.

## 4. Authentic Writing and Reading of Indigenous Children's Picture Books

We introduce the 10 picture books, written by Indigenous authors and in many cases published by Canadian Indigenous publishers, as examples of authentic Indigenous children's literature that can contribute to the promotion of Indigenous children's right to preserve their Indigenous identities (United

Nations, 1989) [43]. In his interpretation of the 10 selected picture books, Red Bear has highlighted the authentic cultural knowledge communicated in the images and in the text of these books [4]. He explains that this knowledge is based on family and community stories shared across generations.

Through reading these picture books with children, teachers can awaken cultural consciousness, leading to respectful relationships amongst all peoples. In the process, the literature contributes to the development of positive Indigenous identities and healing. At a time when authentic and high-quality Indigenous children's literature is more widely available than ever, there is an imperative to include this literature in educational and academic conversations about multicultural children's literature.

We offer Red Bear's interpretations of the 10 picture books, which take up Indigenous perspectives, Indigenous teachings, and his experience, as starting points for teachers' conversations with children when reading the books together. We believe that being a cultural outsider does not preclude non-Indigenous readers from accessing cultural understandings from Indigenous worldviews when reading picture books. Teachers sharing Indigenous books with children have a responsibility to learn about Indigenous cultures from Indigenous peoples, through conversation, lived experience in Indigenous communities, or participating in storytelling, or indirectly through the writings of Indigenous authors, illustrators, publishers, and scholars.

**Author Contributions:** Conceptualization, S.S.P. and R.B.R.; methodology, R.B.R.; writing—original draft preparation, S.S.P. and R.B.R.; writing—review and editing, S.S.P. and R.B.R.; supervision, S.S.P.; project administration, S.S.P.; funding acquisition, S.S.P. All authors have read and agreed to the published version of the manuscript.

**Funding:** This research was funded by the Social Sciences and Humanities Research Council of Canada with a Partnership Grant 410-2010-0431.

**Conflicts of Interest:** The authors declare no conflict of interest.

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
