# Peer review of "Rights of Indigenous Children: Reading Children’s Literature through an Indigenous Knowledges Lens"

_education, doi:10.3390/educsci10100281_

Round 1
Reviewer 1 Report
Rights of Indigenous Children: Reading Children’s 2 Literature through an Indigenous Knowledge Lens
Review
Does the introduction provide sufficient background and include all relevant references?
I thought the authors presented a strong rationale and literature review for this paper.
LL 140 Cite: funds of knowledge (Moll et al.)
Is the research design appropriate?
I wasn’t quite sure what the research design was. I know that the authors were going to read picturebooks from the perspective of an Indigenous Knowledge Keeper, but other than that, I wasn’t quite sure how the authors had determined a design.
Are the methods adequately described?
This section had me somewhat confused. I’m not sure how the authors distilled their selection to 14 books. I saw the four categories, but wanted to know more about the selection of the books, how many were in each category, was there cross-over between and among categories? How was the reading of these books done? How did they develop the categories? Were they generated from the books between 2015-2017 or are these themes that generally run across Indigenous picture books?
I’d like to see the authors discuss critical literacy and critical reader response more fully. As this is the theory through which the literature is read, the readers of this journal would benefit from more explicit explanation of this type of reading. The authors assume that readers of this journal will be familiar with this theory and reader response, but they may not. Further, explaining this theory and then presenting the readings of the picture books would have connected theory and reading more tightly.
Are the results clearly presented?
I’m not quite sure how the authors were presenting their results. From my reading, the authors provide a summary of the different stories within each category, and I wondered how it reflected an Indigenous reading? What aspects of the story retelling were Indigenous (as opposed to non-Indigenous)? I wonder if the authors had considered reading across stories and finding themes within the different groupings of books to talk about the authenticity of the stories presented alongside those interpreted. Presenting a summary of each book was not what I had expected to read. There are glimpses of an Indigenous reading from RB, but more of this would have made the interpretations of the stories clearer.
LL285 and beyond. On line 285, the authors start talking about dreamcatchers without connecting them to a story. The authors talk about dreamcatchers, but I’m not sure to which book the authors are referencing. I started to get lost in the “results section” at this line. I wasn’t sure if this section presented “results” but interpretations of the picture books. Perhaps the authors need to rethink this section to reflect what actually they did with these books. I think the authors could explain some of the misinterpretations of Indigenous objects and symbols in the presentation of the methods would have helped us, the reader, understand how these books represent (or not) accurate Indigenous traditions, symbols, process. I wondered, given that the authors of the picture books are Indigenous, are the authors of this article suggesting that the presentation of information by Indigenous authors/illustrators may be problematic and, perhaps, driven by publishers (e.g., commercialization of dreamcatchers)? I’d like the authors to speak more clearly to the relationship between Indigenous authors/illustrators and the Knowledge Keeper’s interpretations.
Are the conclusions supported by the results?
I wasn’t quite sure of the intention behind the conclusion. The conclusion suggests a parallel with other books that reflect gender, race, disabilities, etc. Books that are written by those/characters involved in stories should have insider knowledge about that which they write. I wanted the authors to say something different. I wanted them to argue for the significance of these insider readings and writings, and how knowledge of Indigenous readings could support educators who teach in Indigenous classrooms as well as those who teach in non-Indigenous classrooms. Conclusions often bring in the theory of reading…how did critical literacy and critical reader response come into play in reading these books? What implications for research, writing, and teaching emerge from this paper?
Author Response
Thank you for identifying the confusion regarding our description of our Indigenous Knowledge approach and the two critical approaches to reading children’s literature. We have described an Indigenous Knowledges approach up front and then contrasted it with the other two critical non-Indigenous approaches that have been used in the literature, although not in our analysis. We ended the Methods section with a description of our specific way of carrying out an Indigenous Knowledges approach. We also provided more information about our selection of the 10 books.
We changed the heading for the interpretations section, cutting the word, “results”, to show that this section reports on RB’s interpretations through an Indigenous Knowledges lens.
We revised the final section to reflect what our paper is about—presenting 10 authentic picture books and the value of RB’s interpretation.
We reference Stolen Words as the book with a dreamcatcher illustration that appears to be a prop. We highlight earlier in the interpretation that the illustrator, Gabrielle Grimard, is non-Indigenous.
We reorganized the introduction to the Intergenerational Impact of Residential Schools section by connecting elements of this theme more closely to the books we interpret. We also create clearer linkages between RB’s interpretations and the themes of each category. We have reduced the categories to three in the process, as it seemed that there was great overlap between two of the categories.
We have removed editorial data from the discussion of the books and have indicated more clearly that much of our descriptions of the books, are indeed, RB’s interpretation, rather than summaries of the books.
Additionally, we made the following revisions:
- Revised the abstract to outline the main findings in the final section.
- Authenticity is defined in section 1 and a clear statement of the specific purpose of this article needs is stated.
- The 3 indigenous publishers are listed chronologically by year of establishment
- Indicated edits have been made.
- We removed the notion of steps to take in the methods section.
- Thanks for checking inconsistencies in number of books analyzed.
- Thank you for identifying the confusion regarding our description of our Indigenous Knowledge approach and the two critical approaches to reading children’s literature. We have described an Indigenous Knowledges approach up front and then contrasted it with the other two approaches. We ended the Methods section with a description of our specific way of carrying out an Indigenous Knowledges approach. We also provided more information about our selection of the 10 books.
- At the start of Section 3 Results, we introduced the four subsections.
- In section 3.1 a reference is provided for the Duncan Campbell Scott quote.
- We acknowledged that Indigenous knowledge and experience are regional, and took up your suggestion about enhancing the validity of our interpretations by interviewing the writers and community members of settings for books outside his home community.
- We explained why we provided information about authors/illustrators for whom we were able to find information. Because we have clarified that we did not use critical methodologies, we have not provided information about which methodology influenced which interpretation.
- We clarified that this article presents authentic cultural ideas and concepts in ten recent picture books, rather than providing evidence for the need for such readings.
Reviewer 2 Report
It was my pleasure to read this paper and I think that its publication will contribute in a significant manner to the state-of-art in the field.
Some typos need to be addressed, e.g. missing spaces and point marks (lines 109, 118, 172).
Regarding the scientific soundness of this paper I would like to advice the authors to look again at the section 3 and try to rewrite some parts in order to increase the support for their arguments.
The authors assume that their paper "present possible cultural readings of selected books". At this point, I am afraid that I am not convinced that they were successful. Therefore, I am suggesting to refine and develop further their analyses - especially in part 3 - in order to fulfill this main objective. The four categories (1.Intergeneration....) need more argumentative developments in their introductory parts. For example, 3.1. Intergenerational impact begins with a paragraph about reconciliation, however no other information is given about their relation, showing clearly how are they related and how they can be developed in reading Indigenous children books. References about specific works in this direction can be added.
Also, I am not fully convinced about the presentation of the literary works included; the inclusion of their full editorial data is disrupting the reading of this paper. My advice is to develop the first part of each category and to show how the category in cause is included in the literary texts. The reading quality of this paper will increase if the author will connect the books presented for each category and develop arguments and connections in this direction.
The approach of the books is not fully developed, especially in part 3.4. My advice is that for each book the authors try to go beyond the mere resume of the books and putt a significant accent on what was announced as possible cultural readings - to fully engage with the books. Otherwise, in some parts it seems that the paper mainly resume some works, without engaging with the text.
Author Response
Thank you for identifying the confusion regarding our description of our Indigenous Knowledge approach and the two critical approaches to reading children’s literature. We have described an Indigenous Knowledges approach up front and then contrasted it with the other two critical non-Indigenous approaches that have been used in the literature, although not in our analysis. We ended the Methods section with a description of our specific way of carrying out an Indigenous Knowledges approach. We also provided more information about our selection of the 10 books.
We changed the heading for the interpretations section, cutting the word, “results”, to show that this section reports on RB’s interpretations through an Indigenous Knowledges lens.
We revised the final section to reflect what our paper is about—presenting 10 authentic picture books and the value of RB’s interpretation.
We reference Stolen Words as the book with a dreamcatcher illustration that appears to be a prop. We highlight earlier in the interpretation that the illustrator, Gabrielle Grimard, is non-Indigenous.
We reorganize the introduction to the Intergenerational Impact of Residential Schools section by connecting elements of this theme more closely to the books we interpret. We also create clearer linkages between RB’s interpretations and the themes of each category. We have reduced the categories to three in the process, as it seemed that there was great overlap between two of the categories.
We have removed editorial data from the discussion of the books and have indicated more clearly that much of our descriptions of the books, are indeed, RB’s interpretation, rather than summaries of the books.
Additionally, we made the following revisions:
- Revised the abstract to outline the main findings in the final section.
- Authenticity is defined in section 1 and a clear statement of the specific purpose of this article needs is stated.
- The 3 indigenous publishers are listed chronologically by year of establishment
- Indicated edits have been made.
- We removed the notion of steps to take in the methods section.
- Thanks for checking inconsistencies in number of books analyzed.
- Thank you for identifying the confusion regarding our description of our Indigenous Knowledge approach and the two critical approaches to reading children’s literature. We have described an Indigenous Knowledges approach up front and then contrasted it with the other two approaches. We ended the Methods section with a description of our specific way of carrying out an Indigenous Knowledges approach. We also provided more information about our selection of the 10 books.
- At the start of Section 3 Results, we introduced the four subsections.
- In section 3.1 a reference is provided for the Duncan Campbell Scott quote.
- We acknowledged that Indigenous knowledge and experience are regional, and took up your suggestion about enhancing the validity of our interpretations by interviewing the writers and community members of settings for books outside his home community.
- We explained why we provided information about authors/illustrators for whom we were able to find information. Because we have clarified that we did not use critical methodologies, we have not provided information about which methodology influenced which interpretation.
- We clarified that this article presents authentic cultural ideas and concepts in ten recent picture books, rather than providing evidence for the need for such readings.
Reviewer 3 Report
The evaluation of children’s literature is important, and cultural contexts are crucial to their evaluation. This article discusses the importance of an Indigenous Literary Aesthetic. The rubrics are clear and the analysis is sound. This is a necessary discussion.
Author Response

(The authors gave the same response as above.)

Reviewer 4 Report
I believe in the thesis of this study: The power of authentic indigenous children’s literature for promoting reconciliation in Canada. I congratulate the authors for embarking on this study. It is important work.
I have suggestions for further developments to tighten the article, presented chronologically by page.
- A last sentence is needed for the abstract to outline the main findings.
- In the first section, a clear and convincing explanation is given for the value of indigenous children’s literature for both indigenous and non indigenous readers.
- Authenticity is central to a lot of this article. It should be defined in section 1.
- A clear statement of the specific purpose of this article needs to be stated in section 1. To me it is about assessing the authenticity of ten recent indigenous English language Canadian published picturebooks.
- Minor note: P2 para2 suggest listing the three indigenous publishers chronologically by year of establishment (i.e., Theytus, Kegedonc, Inhabit)
- Line 109 full stop missing after [16,17]
- Line 137 Please clarify ‘The first step’ of what? If this is a method of a certain number of steps, please outline what these are to orientate the reader.
- 4 Materials and Methods. Please clarify how many books were analysed. In 2.1 ten books are mentioned. In 2.2 fourteen books are mentioned. The results section would suggest the final number was ten.
- In Materials and Methods where the books are taken from is clearly explained, but how the books are selected is not explained. Are the 10 or 14 books all of the books from the lists described, or a selection? If the latter, how were they chosen? What criteria were used?
- In section 2.2 I become confused about the thee approaches used for analysis. Are they (1) Indigenous Knowledge Approach, (2) Critical Literacy Approach, (3) Critical Readers’ response approach? This could be clarified in the text.
- At the end of Section 2, an outline of how these analytical approaches were applied is needed.
- At the start of Section 3 Results, a sentence to introduce the four subsections could be added.
- In section 3.1 a reference is needed for the Duncan Campbell Scott quote.
- I really appreciated the links made to the indigenous knowledge of RB throughout your analysis of the ten books in section 3. One question I had was, because indigenous knowledge and experience is regional, might it have been a good idea to only analyse books from RB’s regional affiliation; or maybe to include other knowledge keepers related to the regions linked to each book. Another approach (and this is probably a future study) would be to interview the writers of each other books.
- I like the way the affiliations of each author is given. For me, the analysis of the books could be presented more methodically. I would like to see affiliations (or not) of all authors and illustrators given, and some analysis of illustrations given for each book, even if to say the illustrations appear authentic in all ways. I would also like more clear indexing of the three methods mentioned earlier. When is the analysis as the result of approaches (1), (2) or (3)?
- It seems to me that this article explores how authentic the cultural ideas and concepts in ten recent picturebooks are. At this stage the conclusion could be tightened to reflect this main idea.
- With respect to this line in the last paragraph: “Additionally, as we have shown in our paper, the ways in which books by Indigenous authors, illustrators and publishers are read and discussed should be aligned with holistic Indigenous knowledges and perspectives”, I am not sure that this article does show this. It shows to some extent how they are aligned, but not how they ‘should be’ aligned.
I thank the authors for the work and thought that has gone into the presentation of this important article. I truly believe in the importance of the work presented here. I hope my suggestions are of help; my apologies if any of them represent misunderstandings on my part.
Author Response

(The authors gave the same response as above.)

Round 2
Reviewer 1 Report
See attached file

Author Response
As we explained in our previous list of revisions, we are not using critical theory or any of the other methods that have typically been used to analyze children’s literature. We have described an Indigenous Knowledges approach up front and then contrasted it with the other two critical non-Indigenous approaches that have been used in the literature, although not in our analysis. We ended the Methods section with a description of our specific way of carrying out an Indigenous Knowledges approach. Critical literacy is not our methodology so there is no need, in our opinion, to give it more than a mention as a more widely-practiced method in literature analysis. We also provided more information about our selection of the 10 books, as requested by the reviewer.
Although the reviewer perceives our analysis as a summary of the books, we assure the reviewer that we did not summarize the books. We only highlighted the characteristics of the narrative and of the illustrations that Red Bear, in his analysis using his perspective as an Indigenous Knowledge Keeper. The dreamcatcher in the illustration, for example, is highlighted as a settler’s stereotype of Indigenous culture. A settler reading this book might not notice how jarring it is that the illustration contains the dreamcatcher. An Indigenous reading is important to highlight the stereotypes in illustrations created by settlers. We reference Stolen Words as the book with a dreamcatcher illustration that appears to be a prop. We highlight earlier in the interpretation that the illustrator, Gabrielle Grimard, is non-Indigenous.
We changed the heading for the interpretations section, cutting the word, “results”, to show that this section reports on RB’s interpretations through an Indigenous Knowledges lens. At the start of Section 3 Results, we introduced the four subsections.
We attended to the reviewer’s request for changes to the concluding section. As shown in the yellow highlights, this section has been completely revised to reflect what our paper is about—presenting 10 authentic picture books and the value of RB’s interpretation. We clarified that this article presents authentic cultural ideas and concepts in ten recent picture books, rather than providing evidence for the need for such readings.